Non-agriculturalization of cultivated land in densely populated areas at the watershed scale: a case study of the Minjiang River Basin, China

Zhao Xi 1 2
Hu Zhongwen zwhoo@szu.edu.cn 1 2
Zhang Yinghui 1 2
Wang Jingzhe 3
Shi Tiezhu 1 2
Liu Yanguo 4
Zhang Jie 5
Wu Guofeng 1 2
1 School of Architecture and Urban Planning, Shenzhen University , Shenzhen , China
2 MNR Key Laboratory for Geo-Environmental Monitoring of Great Bay Area & Guangdong Key Laboratory of Urban Informatics & Shenzhen Key Laboratory of Spatial Smart Sensing and Services, Shenzhen University , Shenzhen , China
3 School of Artificial Intelligence, Shenzhen Polytechnic University , Shenzhen , China
4 Key Laboratory of Investigation, Monitoring, Protection and Utilization for Cultivated Land Resources, Ministry of Natural Resources , Chengdu , China
5 College of Information and Electrical Engineering, China Agricultural University , Beijing , China
Sunny Armando
Electronic publication date: 2025 Jul 22
Publication date: 2025
Volume: 13
Electronic Location ID: e19722
Received 2025 Feb 24; Accepted 2025 Jun 17
Copyright: ©2025 Zhao et al.
Copyright year: 2025
Copyright holder: Zhao et al.
License: This is an open access article distributed under the terms of the Creative Commons Attribution License, which permits unrestricted use, distribution, reproduction and adaptation in any medium and for any purpose provided that it is properly attributed. For attribution, the original author(s), title, publication source (PeerJ) and either DOI or URL of the article must be cited.
License URL: https://creativecommons.org/licenses/by/4.0/

Keywords: Sustainable development goals, Zero hunger, Food security, Non-agriculturalization of cultivated land, Apatiotemporal analysis

Funding: Shenzhen Science and Technology Program JCYJ20220818101617037 JCJY20230808105201004 National Natural Science Foundation of China 42471351 42201347 This work was supported by Shenzhen Science and Technology Program (JCYJ20220818101617037, JCJY20230808105201004), the National Natural Science Foundation of China (42471351, 42201347). The funders had no role in study design, data collection and analysis, decision to publish, or preparation of the manuscript.

==============================
Zero hunger is a top priority in the Sustainable Development Goals, however, millions of people worldwide still face hunger. Over the years, China has experienced rapid population growth, industrialization and urbanization, leading to significant agricultural land loss, which threats the nation’s food supply. Understanding the patterns and driving factors of non-agriculturalization is crucial for its protection. The Minjiang River Basin, the largest tributary of the Yangtze River, is densely populated and experiencing rapid urbanization, making it a typical snapshot of the non-agriculturalization in China. This study comprehensively examines the characteristics and trends in the spatiotemporal evolution of cultivated land in the Minjiang River Basin, along with the drivers of non-agriculturalization. The results reveal the findings: (1) From 1990 to 2020, the cultivated land in the Minjiang River Basin has decreased and mainly concentrated in flat areas. The spatiotemporal evolution of cultivated land exhibits low dispersion and strong directionality, predominantly shifting northward, and the migration of cultivated land can be divided into three stages. (2) A consistent positive spatial correlation is observed in the non-agriculturalization areas of cultivated land in the Minjiang River Basin, with high-high (HH) clustering analysis revealing an aggregation pattern radiating outward from the city center. (3) Population growth and economic factors are the primary drivers of non-agriculturalization in the Minjiang River Basin.

Introduction

The United Nations’ 2015 Sustainable Development Agenda identified zero hunger as a key goal among the 17 Sustainable Development Goals (SDGs) to be achieved by 2030. However, approximately 9.2 percent of the global population faced chronic hunger in 2022, up from 7.9 percent in 2019. Current projections suggest that nearly 600 million people will be chronically undernourished in 2030, underscoring the immense challenge of achieving the SDG target to eradicate hunger (FAO, 2023).

Cultivated land plays a fundamental role in national food security. Since the reform and opening of the country, China’s rapid industrialization, urbanization, and population growth have exacerbated land scarcity. In 2022, China’s cultivated land covered 1.276 million km2, with a per capita area of 0.09 hectares, half the global average of 0.18 hectares. The phenomenon of non-agriculturalization, defined as the conversion of farmland for urban and industrial purposes, has notably diminished cultivated land. This trend undermines the objective of maintaining 1.2 million km2 of farmland and ensuring food security. Consequently, there is an urgent need to monitor the non-agriculturalization process and understanding its driving factors, which is crucial for developing more effective management and protection policies to ensure the sustainable development (Liu, Zhao & Song, 2017; Huang et al., 2017; Chen & Yao, 2024).

Many researchers have concentrated on various aspects such as the spatiotemporal characteristics (Wang, Li & Luo, 2020; Li et al., 2022; Gao et al., 2025; Xiong et al., 2025), influencing factors such as the economy and transportation (Zhang et al., 2018; Zhang et al., 2024; Li et al., 2021; Jiang et al., 2023; Wu et al., 2024), and regulatory instruments (Potapov et al., 2022; Xiandong et al., 2022; Qie et al., 2023; Chen & Yao, 2024; Zeng et al., 2025) related to the non-agriculturalization of cultivated land. Economic and population factors are commonly regarded as the primary drivers of this phenomenon (Chaudhary et al., 2020; Yang et al., 2021; Li, Huang & Han, 2022; Chen, Wang & Wang, 2022; Chen, Li & Xia, 2022). Recent studies have focused on changes in China’s cultivated land, particularly concerning land protection and food security policies, due to the country’s substantial population and high food demand (Liu, Zhao & Song, 2017; Huang et al., 2017; Zhou, Li & Liu, 2021; Chen et al., 2024). It is suggested that certain regions in China have disproportionately utilized agricultural land to facilitate economic development. While numerous studies have investigated large-scale regions concerning the non-agriculturalization of cultivated land (Huang, Hou & Yao, 2022; Wang & Zhang, 2023), studies on cultivated land dynamics at local scales is crucial in helping stakeholders understanding the factors driving changes and identify viable land management strategies (Zhang et al., 2018; Gao et al., 2025).

Sichuan Province, renowned as a major grain-producing region in China, boasts extensive cultivated land and a high population density (Zhang & Dong, 2023), and the Minjiang River Basin is the major grain-producing area in Sichuan Province. However, the region faces challenges such as dwindling total cultivated land area, deteriorating land quality, and low productivity, which pose significant obstacles to cultivated land protection. The basin is also grappling with limited cultivated land resources due to a high population density, uneven water resource distribution, imbalanced economic development, and significant human-land conflicts (Huang & Xu, 2023). The total cultivated land in the Minjiang River Basin stood at a mere 13,133.38 km2 in 2020, with a per capita share of 0.05 hectares, significantly below the national standard. Consequently, the non-agriculturalization of the Minjiang River Basin is a typical snapshot of China’s non-agriculturalization process, particularly in high-density population areas.

However, the process, patterns and driving factors of non-agriculturalization in the Minjiang River basin remain insufficiently understood. Therefore, this study aims to address these gapes by utilizing remote sensing data, GIS techniques, and geographical detector, focus on three key aspects: (1) evaluating the spatial distribution and changes in cultivated land within this basin; (2) providing a detailed quantification of the spatiotemporal patterns of non-agriculturalization; and (3) analyzing the driving factors of non-agriculturalization by quantifying the effects of natural, socioeconomic, and demographic factors. Our findings are anticipated to offer fresh insights into sustainable development in this basin, as well as some similar densely populated areas.

Study Area and Materials

Study area

The location of the study area is presented in Fig. 1. The Minjiang River (103°3′E–104°40′E, 28°25′N–31°2N), situated in the southeast of Sichuan Province, China, boasts a basin characterized by low-lying terrain and a monsoon climate, experiencing substantial annual precipitation, with averages ranging from 1,000 to 1,600 mm. The Minjiang River Basin spans a total area of 22,051.09 km2. In 2020, cultivated land occupied 13,133.38 km2, accounting for approximately 59.56% of the entire middle and lower river area. Originating from Dujiangyan, the Minjiang River flows southward and merges with the Yangtze River at Yibin, passing through basins and hilly regions. The primary crops in the Minjiang River Basin are rice, corn, wheat, soybeans, collectively earning it the esteemed title of the “granary of Sichuan Province”. However, over the past 30 years, urbanization and tourism resource development have been vigorously pursued in the basin to promote local economic growth. This has inevitably resulted in negative impacts such as unreasonable allocation of resources and environmental degradation, with a particularly acute impact on land resources (Yao et al., 2023).

Figure 1 Location of the study area.

Materials

This study analyzed the characteristics and trends of non-agricultural land use within the area using land use data derived from remote sensing images. Furthermore, investigated the drivers of non-agriculturalization by integrating elevation data, statistical yearbooks, and water resources bulletin.

The land use data for 1990, 1995, 2000, 2005, 2010, 2015, and 2020 in the Minjiang River Basin were sourced from The China Multiperiod Land Use Land Cover Change (CNLUCC) remote sensing monitoring dataset, available through the Resource and Environment Science Data Registration and Publication System (https://www.resdc.cn). It is a national-level multi-temporal land-use thematic database constructed using Landsat remote sensing images as the main information source, with a spatial resolution of 30 m and an accuracy up to 95%.

To explore the drivers of non-agriculturalization, digital elevation model (DEM) data was obtained from the Google Earth Engine (GEE) platform, with a spatial resolution of 90 m. The slope and aspect data within the basin were both extracted based on this elevation data. In addition, highway mileage, average temperature, precipitation, average relative humidity, sunshine duration, gross regional product, primary industry, secondary industry, tertiary industry, proportion, total resident population, rural population, urban population, population density of the study area were all obtained from the “Sichuan Statistical Yearbook” and various city statistical yearbooks. The precipitation in certain areas is sourced from the local water resources bulletin. However, due to the unavailability of statistical data of several cities (include Liangshan, Meishan, Neijiang, Ya’an, Yibin, and Zigong), the driving factors from 1990 to 1995 could not be conducted at this time.

Methods

This study examined the degree of aggregation and the spatiotemporal evolution trends of cultivated land, along with the non-agricultural use of cultivated land, utilizing a variety of methods. Moreover, it delved into the underlying driving factors contributing to the non-agriculturalization of cultivated land. The technical framework is delineated as follows (Fig. 2):

Figure 2 Technical framework.

Agglomeration analysis of agricultural land

To investigate the agglomeration of agricultural land in the Minjiang River Basin from 1990 to 2020, the kernel density estimation (KDE) is used. KDE is an exploratory statistical method that measures the density and the spatial distribution of cultivated land use (Terrell & Scott, 1992). KDE functions by placing a kernel at each data point and summing them to produce a continuous probability density curve (Majdara & Nooshabadi, 2020).

In KDE analysis of cultivated land, a higher density signifies a more concentrated distribution of cultivated land, whereas a lower density implies a more dispersed layout. Regarding the KDE of non-agriculturalization on cultivated land, a higher density indicates a more significant non-agriculturalization trend, while a lower density suggests a lesser conversion of cultivated land into other land types.

Spatial pattern and migration trend of cultivated land

The standard deviational ellipse (SDE) is employed to analyze the direction and trend of spatial distribution, reflecting the predominant distribution direction of agricultural land and the degree of dispersion in each direction. The parameters of a standard deviation ellipse comprise the X and Y coordinates of the mean center, the major and minor axes of the ellipse, and the orientation of the ellipse. The calculation formula and detailed methods are described in the literature (Lefever, 1926; Wang et al., 2025; Wu et al., 2025).

In the SDE analysis of cultivated land, the long axis signifies the distribution direction, and the short axis represents the distribution range. The center of the ellipse indicates the average density center of the cultivated land. We performed SDE analysis of cultivated land in the Minjiang River Basin over various periods to investigate the changing spatial patterns of cultivated land throughout the years.

Spatial pattern analysis of non-agriculturalization

In this study, the CNLUCC dataset classifies land use into six primary categories: cropland, woodland, grassland, water, urban, and barren soil. For the purposes of this study, the conversion of cropland to other land types is termed non-agriculturalization of cultivated land.

To explore the spatial correlation and spatial differences in non-agriculturalization areas within the basin, global and local spatial autocorrelation analysis methods were utilized.

Global spatial autocorrelation can analyze the spatial relationship of features across the entire basin. Additionally, Global Moran’s I was used in this study to measure the global autocorrelation (Moran, 1950). The value of Moran’s I ranges from 0 to 1. A value greater than 0 indicates a positive correlation between spatial features, with higher values signifying stronger clustering. A value of 0 indicates no significant correlation, while a value less than 0 indicates a negative correlation, with smaller values denoting more significant negative correlation and greater spatial differences.

Local spatial autocorrelation can effectively reveal the degree of difference and significance level between each spatial unit and its neighboring units (Anselin, 1995). Therefore, this study used Local Moran’s I to measure local spatial autocorrelation. Local spatial autocorrelation categorizes the spatial attribute data of features into high-high (HH), low-high (LH), low-low (LL), high-low (HL), and non-significant (NN) types. HH indicates that the attribute values of a unit and its surrounding units are all relatively high, while HL indicates that the attribute value of a unit is high, but the attribute values of surrounding units are low.

The Z-score was used to test the significance level of global and local spatial autocorrelation of non-agriculturalization in the basin (Zhu et al., 2020). A Z-score greater than 1.96 or less than −1.96 indicates significant spatial autocorrelation of non-agriculturalization areas within the spatial unit.

Geographical detector

Exploring the driving mechanisms behind non-agriculturalization is beneficial for a deeper understanding of the essence of non-agriculturalization and can provide theoretical basis for the scientific and reasonable allocation of land resources by regional departments (Wang & Xu, 2017). Therefore, the geographical detector is employed in this study to explore the drivers of non-agriculturalization. The geographical detector is a statistical method used to detect spatial differentiation and driving factors. Its basic idea is to divide the study area into several sub-regions. If the sum of variances in the sub-regions is smaller than the total variance of the region, there is spatial differentiation. If the spatial distributions of two variables tend to be consistent, there is a statistical correlation between them. Two detectors were used in this study:

(1) Factor detector: this detector assesses the spatial differentiation of non-agriculturalization area values and the explanatory power of factor X values for non-agriculturalization area Y values.

(2) Interaction detector: this detector examines whether the explanatory power of factor Xn and Xn+1 on Y values is affected when they act together.

Results and Analysis

Spatiotemporal evolution analysis of cultivated land

Aggregation degree analysis of cultivated land

This study conducted a KDE of cultivated land in the Minjiang River Basin for the year 1990, and used Jenks optimization to classify the cultivated land density into five levels. KDE is effective in performing spatial smoothing on land use changes, integrating originally discrete and discontinuous land use information (Brandes et al., 2024). The division into five levels helps to more accurately capture the characteristics of cultivated land changes across different periods in the region. The classification from high-value areas to low-value areas clearly displays the distribution pattern and changing trends of cultivated land. Subsequent analysis was conducted based on this classification, with results depicted in Fig. 3 and density values detailed in Table 1.

As shown in Fig. 3, the spatiotemporal distribution pattern of cultivated land in the Minjiang River from 1990 to 2020 shows a relatively consistent trend. Cultivated land demonstrates dense concentrations in the central and southeastern regions, while being sparsely distributed in the southwestern and northwestern regions. The high-value areas of cultivated land are mainly located in the flat areas in the central and northeastern parts of the Minjiang River Basin, as well as in the southeastern regions. The median-value areas of cultivated land distributed in various regions, while the low-value areas are primarily concentrated in the southwestern and northwestern regions of the study area. The results from 1990 to 2010 show the most obvious changes in cropland centered in the main urban area of Chengdu City, which has been declining for many years. From 2010 to 2015, cropland in Leshan City’s center experienced significant reduction.

Over the past 30 years, the total area of cultivated land in the Minjiang River Basin has declined from 14,706.18 km2 to 13,133.38 km2, indicating a declining trend. The average value of the density shows a decreasing-increasing-decreasing trend, as shown in Table 1. However, from 1990 to 2020, all cities and counties have experienced different degrees of non-agriculturalization. A total of 4,308,179 plots of cultivated land were reduced from 1990 to 2020, with the average value decreasing from 729 to 651 plots per square kilometer. The most pronounced reduction occurred between 2010 and 2015, with low-value cultivated land areas expanding from the periphery towards central regions, notably around the Chengdu urban core.

Figure 3 The aggregation degree of cultivated land.

Table 1 Distribution density values of cultivated land plots.

	1990	1995	2000	2005	2010	2015	2020	
Aggregate	40,157,860	38,696,062	39,423,428	40,580,252	39,486,522	36,881,237	35,849,681	
Average	729	702	716	732	717	670	651	
Notes.

“Aggregate” refers to the total number of plots, while the unit of “average” is expressed as plots per square kilometer.

Spatial pattern and migration trend of cultivated land

This study utilized SDE to examine the spatial pattern and distribution trend of cultivated land in the Minjiang River Basin from 1990 to 2020. The distribution of cultivated land exhibited a distinctive north-south concentration, forming an ellipse stretching from the southern city of Yibin to the northern city of Chengdu. The closer the axis ratio of the ellipse is to 1, the more circular the spatial distribution, indicating greater dispersion and weaker spatial distribution directionality. The average ratio, as seen in Table 2, was 0.377, indicating a relatively low degree of dispersion and strong spatial directionality. The ratio remained consistent over the past 30 years, with a ratio of 0.36 in 2005, signifying the strongest spatial clustering of non-agricultural cultivated land during this period.

Table 2 Attributes of the standard deviation ellipse for cultivated land.

	Standard deviation of the major axis	Standard deviation of the major axis	Central coordinates	Axis ratio	
1990	106,501.93	44,217.34	29°28′48″N, 103°48′44″E	0.42	
1995	110,793.02	39,451.13	29°34′14″N, 103°45′51″E	0.36	
2000	112,807.27	43,013.45	29°33′29″N, 103°48′55″E	0.38	
2005	121,953.49	44,472.10	29°41′04″N, 103°45′58″E	0.36	
2010	119,379.96	440,35.97	29°41′10″N, 103°47′41″E	0.37	
2015	119,422.45	43,864.60	29°44′24″N, 103°46′21″E	0.37	
2020	118,852.37	45,072.74	29°44′47″N, 103°48′22″E	0.38	

As shown in Fig. 4A, from 1990 to 2020, the cultivated land in the Minjiang River Basin primarily exhibited a northward movement trend, characterized by a “fast-slow” cycle every 10 years. The most rapid center movement of the cultivated land center occurred between 2000 and 2005, with a peak rate of 2.96 km/year. When analyzing the results of the SDE that illustrate the northward migration of cultivated land (Table 3), it is essential to consider the roles of topography and policy. Despite the predominantly flat topography of the area, except for certain sections, the evolution of cultivated land has been minimally impacted by the terrain. In terms of policy, rapid urbanization and industrialization in Leshan City, particularly represented by the Lingang Economic Development Zone construction and the “Transportation Offensive Campaign,” have led to a decline in cultivated land in the southern region, prompting the centroid’s northward shift. The major axis and area of the ellipse serve as indicators of cultivated land dispersion, with longer axes and larger areas denoting increased dispersion. Additionally, as depicted in Fig. 3, cultivated land has diminished across diverse regions, indicating a rise in distribution dispersion over time.

Figure 4 Migration trend of cultivated land.

(A) Migration trend of the elliptical center of cultivated land and (B) migration trend of the elliptical of cultivated land.

As shown in Fig. 4B, the migration of the SDE can be divided into three stages:

(1) 1990 to 2000: the long axis spread in the northwest direction, while the short axis initially shrank and then spread expanded, signifying a northwest migration of cultivated land that first clustered and then dispersed.

(2) 2000 to 2005: during this period, the center of cultivated land moved at the fastest rate, with a notable increase in the major axis of the ellipse and an expansion of the cultivated land area.

(3) 2005 to 2020: the major axis showed a “shrinkage-expansion-shrinkage” pattern, indicating that the cultivated land initially clustered towards the center, then dispersed towards the north and south, and finally clustered again. The minor axis demonstrated an initial contraction followed by expansion, indicating a convergence and subsequent dispersion of cultivated land.

Spatiotemporal evolution analysis of non-agriculturalization

Quantity analysis of non-agriculturalization

With a 5-year change cycle, the land use transfer matrix was employed to analyze the changes in land use from 1990 to 2020. As shown in Fig. 5, the proportion of non-agriculturalization in the Minjiang River Basin exhibited a fluctuating trend, which initially declined and then raised from 1990 to 2020. Notably, the most common conversion of cultivated land was to woodland, followed by construction land. The most intense stage of non-agriculturalization occurred from 2010 to 2015, reaching a peak of 9.949%.

Table 3 The parameters of the standard deviation ellipse for cultivated land.

	Centroid
displacement	Migration
rate	Major axis variation	Minor axis variation	
1990∼1995	11.03	2.21	4,291.09	−4,766.21	
1995∼2000	5.13	1.03	2014.24	3,562.32	
2000∼2005	14.80	2.96	9,146.22	1,458.65	
2005∼2010	2.75	0.55	−2,573.53	−436.13	
2010∼2015	6.37	1.27	42.49	−171.37	
2015∼2020	3.29	0.66	−570.08	1,208.14	

Figure 5 Land use change pattern in the Minjiang River Basin from 1990 to 2020.

Spatial aggregation degree analysis of non-agriculturalization

Focusing on the most intense period of non-agriculturalization from 2010 to 2015, this study conducted a KDE of non-agricultural land in the Minjiang River Basin. The density was divided into eight levels using the Jenks optimization. In the color scale, the colors correspond to the non-agrochemical density grades, which range from low to high as follows: 0–30, 31–76, 77–136, 137–208, 209–299, 300–462, 463–685, and 686–965 plots per square kilometer. The greater the value, the more intense the phenomenon of non-agriculturalization. Subsequent stages were analyzed based on this classification, with results depicted in Fig. 6.

Figure 6 Spatial aggregation degree of non-agriculturalization.

As shown in Fig. 6, the most intense non-agriculturalization during 1990–2020 occurred in the northern part of Chengdu, showing a trend of outward expansion from the center of the main urban area. High-value non-agriculturalization areas in other regions mainly represented dispersed areas expanding from administrative boundaries.

From 1990 to 1995, intensive non-agriculturalization occurred in the northwest region, with partial non-agriculturalization observed in the southeastern region, along the eastern boundary, and in the central-southern region. From 1995 to 2000, 2000 to 2005, and 2005 to 2010, intense non-agriculturalization was observed in the northeast region, with partial non-agriculturalization in the county centers of the central and southeastern regions. The peak period of 2010 to 2015 witnessed the most intense non-agriculturalization over the 30-year span, particularly in the northern region. However, Chengdu and Dujiangyan have relatively lower levels of non-agriculturalization than other areas. Lastly, from 2015 to 2020, intense non-agriculturalization occurred in the northwest region, with scattered areas of relatively low-level non-agriculturalization in other regions.

The spatial distribution pattern of non-agriculturalization

Based on the size of the study area, this study divided the Minjiang River Basin into grids with a scale of 10,000 × 10,000 m and conducted global and local spatial autocorrelation. The results, presented in Table 4, demonstrated that the Moran’s I for the non-agriculturalization area of 1990–2020 ranged from 0.33 to 0.71, with P-values less than 0.001 and Z-values greater than 2.58, indicating a positive spatial correlation in the distribution of non-agriculturalization areas, with an initial decrease, followed by an increase and then a subsequent decrease in aggregation.

Table 4 Moran’s I of non-agriculturalization.

	Moran’s I	Z value	P value	
1990∼1995	0.71	21.19	<0.001	
1995∼2000	0.69	20.57	<0.001	
2000∼2005	0.34	10.09	<0.001	
2005∼2010	0.33	9.74	<0.001	
2010∼2015	0.48	14.48	<0.001	
2015∼2020	0.46	13.76	<0.001	

During 1990–1995 and 1995–2000, the Moran’s I values were significantly higher than those of the other four periods, indicating that the aggregation of non-agriculturalization areas in the Minjiang River Basin was significantly greater during 1990–2000 compared to the period of 2000–2020.

The spatial distribution pattern is detailed in Fig. 7. From 1990 to 1995, high-high (HH) clusters were predominantly concentrated in the northwest region of the middle and lower reaches of the Minjiang River Basin, specifically in Chengdu City, Meishan City, and Leshan City. Conversely, low-low (LL) clusters were primarily found in the northeast and southwest regions. Transitioning to 1995 and 2000, HH clusters shifted to the south-central region of the Minjiang River Basin, encompassing Leshan City, Meishan City, Neijiang City, Zigong City, and Yibin City. LL clusters, on the other hand, were more prevalent in the northwest and southeast regions during this period. Progressing to 2000–2005, HH clusters exhibited a more dispersed pattern, particularly in the western part of the Minjiang River Basin, covering a wider geographical area compared to the previous decade. Notably, HH clusters were concentrated in Chengdu City, Ya’an City, Meishan City, Leshan City, and Yibin City in the southwestern region. LL clusters remained concentrated in the southwestern part of the basin. Moving forward to the period between 2005 and 2010, HH clustering was predominantly observed in the south-central part of the Minjiang River Basin, while LL clustering was limited to specific areas in the northwest and southwest. The HH clusters once again became concentrated in the central part of the basin, including Leshan City, Meishan City, and Yibin City, whereas LL clusters were situated in the northwestern and southwestern regions. Between 2010 and 2015, HH clustering was primarily concentrated in the west-central part of the Minjiang River Basin, while LL clustering was more prevalent in the southwest region. LL clusters were mainly distributed in the southwest and northeast regions, with Chengdu City, Ya’an City, and Leshan City identified as HH clusters. In the final period of 2015–2020, HH clusters continued to be concentrated in the central and western parts of the Middle and Lower Minjiang River Basin, with a broader distribution compared to the previous five-year period. LL clusters were predominantly located in the southwestern and northeastern regions of the basin.

Figure 7 Spatial aggregation degree of non-agriculturalization.

Drivers of non-agriculturalization

In this study, a geographic detector was employed to scrutinize the influence of various driving factors of the non-agriculturalization in the Minjiang River Basin from 1995 to 2020. Due to the lack of statistical data, the driving factors from 1990 to 1995 could not be analyzed in this study. As shown in Table 5, seventeen indicators were selected as driving factors, including terrain, transport, climate, economy, and population, and they are categorized into five types. The area of non-agriculturalization was considered as the dependent variable (y).

Table 5 Factor selection.

Factor	Number	Driving factors	Importance	
Terrain	X1	Elevation	There is an abundance of cultivated land in flat regions, making development relatively straightforward; however, development in high-altitude areas poses significant challenges	
X2	Slope	The steep slope complicates mechanization, reduces agricultural production efficiency, and increases the likelihood of cultivated land being abandoned or repurposed for other uses	
X3	Aspect	Slope orientation affects the growth cycle of crops and, consequently, influences agricultural yields. Low agricultural efficiency may prompt the conversion of land to alternative uses	
Transport	X4	Highway mileage	Convenient transportation will enhance population mobility and stimulate economic development, resulting in the utilization of cultivated land	
Climate	X5	Average temperature	Crops require the appropriate temperature for optimal growth, which in turn affects agricultural yields. Low agricultural efficiency may lead to the conversion of farmland for alternative uses	
X6	Precipitation	Precipitation is essential for crop growth and significantly impacts agricultural yields	
X7	Average relative humidity	Humidity significantly affects crop growth, which in turn impacts agricultural yields	
X8	Sunshine duration	Sufficient sunlight is a key factor in determining crop growth trends, which in turn influences agricultural yields	
Economy	X9	Gross regional product	An important indicator of social and economic development is the conversion of cultivated land into industrial, commercial, and urban construction land	
X10	Primary industry	Agriculture (planting, forestry, animal husbandry, fishing, etc.) directly demonstrates the utilization patterns of cultivated land	
X11	Secondary industry	Industry (extractive, manufacturing, etc.) Manufacturing, production and supply of electricity, gas and water) and construction, will directly impact cultivated land	
X12	Tertiary industry	Other industries, aside from primary and secondary sectors, will directly occupy cultivated land	
X16	Proportion	An important indicator of the level of urbanization is the migration of the rural population to cities	
Population	X13	Total resident population	The change in population size exerts pressure on land resources	
X14	Rural population	The decline in the rural population may result in the abandonment of cultivated land, which can subsequently be repurposed for non-agricultural purposes	
X15	Urban population	The growth of the urban population will lead to an expansion of urban development, encroaching upon cultivated land	
X17	Population density	Population density influences the distribution of the rural labor force and the intensity of land use	

As shown in Fig. 8, the results indicate that economic and population factors are the primary driving forces behind the non-agriculturalization from 1990 to 2020, contrasting with the relatively lower influence of terrain, transportation, and climate factors.

Figure 8 Effects of different factors on Non-Agriculturalization in the Minjiang River Basin from 1995 to 2020.

From 1995 to 2000, population emerged as the primary driver, with total population exhibiting 90% explanatory power. From 2000 to 2005, transportation, population, and economic development became the main driving factors, with the length of highways having the highest explanatory power at 95%. Compared to the period of 1995–2005, the explanatory power of each factor for non-agriculturalization decreased overall after 2005. During the periods 2005–2010, 2010–2015, and 2015–2020, transportation and population remained the main driving factors, but the highest explanatory power for a single factor was only around 50%. The q-value of certain driving factors from 2005 to 2020 ranges between 0.4 and 0.6. This may be attributed to the rapid and complex socioeconomic changes that China experienced during this period. For instance, the acceleration of urbanization was accompanied by adjustments in industrial structures. The development of emerging industries resulted in a more diverse demand for land, complicating the impact of traditional driving factors on non-agriculturalization.

A single factor can only partially explain the drivers of non-agriculturalization, making it necessary to calculate the results of multiple-factor interactions for auxiliary verification. The findings indicate that the interaction of multiple factors exhibits non-linear enhancement and dual-factor enhancement. Figure 9 demonstrates that the combined effect of multiple factors on non-agriculturalization in the Minjiang River Basin is greater than the explanatory capacity of any single factor.

Figure 9 Effects of the interaction of different factors on the non-agricultural conversion of cultivated land in the Minjiang River Basin from 1990 to 2020.

During 1995 to 2000, the population density factor X17 exhibited an explanatory power exceeding 90% when interacting with various other factors. Notably, interactions with X11 and X9 reached up to 99%, indicating a strong influence. Additionally, the combination of gross regional product (GRP) and the secondary industry underscored the pivotal role of economic factors in the non-agriculturalization process.

In the period from 2000 to 2005, a multitude of factors including X4, X10, X11, X12, X13, X14, X15, X16, and X17, showed high explanatory power, predominantly exceeding 90%. The collaborative efforts of transportation, economic, and demographic factors were instrumental in driving the non-agriculturalization of cultivated land. Subsequently, from 2005 to 2010, there was a decline in the overall explanatory power of factor interactions compared to the preceding periods, with only X4 and X14 exhibiting an explanatory power of 91.2%.

During 2010 to 2015, the interaction between X16 and factors X13, as well as X16 and X15, demonstrated explanatory powers of 92.8% and 97.6%, respectively. Notably, from 2015 to 2020, no factor interactions exceeded 90% explanatory power, with the highest recorded at 88.9% for X4 and X15. Particularly, the interaction involving factor X4 with other factors showed the most significant improvement in explanatory power.

To summarize, the collective impact of transportation, population, and economic factors significantly influenced the non-agriculturalization of cultivated land in this study area.

Discussion

Large amounts of cultivated land converted into construction land and forest

Over the past 30 years, the total amount of cultivated land in the Minjiang River Basin has decreased from 14,706.18 km2 in 1990 to 13,133.38 km2 in 2020. A substantial portion of cultivated land has been converted into forest land and construction land due to the implementation of urbanization processes and policies for returning cultivated land to forests.

During 1990–2020, the conversion of cultivated land to construction land in the Min River Basin was particularly prominent. As shown in Fig. 5, a total of 914.25 km2 of cultivated land was converted into construction land, while the reverse conversion accounted for only 3.63 km2, resulting in a net loss of 910.62 km2. Among these changes, the Chengdu City, as a typical region for non-agriculturalization, demonstrates a highly representative trend (Fig. 6). Located at the core of the Chengdu Plain, Chengdu has leveraged its favorable natural and geographical conditions to rapidly rise as a key engine of economic development in western China since the launch of the Western Development Strategy in 2002. From 2000 to 2020, Chengdu’s Gross Domestic Product (GDP) soared from 123.82 billion yuan to 1,771.67 billion yuan. However, this rapid economic growth has also driven significant land demand. Data show that in 1990, Chengdu had 9,909.44 km2 of cultivated land and a total grain output of 3.817 million tons; by 2020, the cultivated land area had sharply decreased to 7,288.70 km2, and total grain output had fallen to 2.2786 million tons, highlighting the encroachment of urban construction on farmland resources.

In terms of the conversion between cultivated land and forest land, a total of 3,900.73 km2 of cultivated land in the Min River Basin was converted into forest land over the past 30 years, while 3,281.58 km2 of forest land was conversely converted back into cultivated land, resulting in a net loss of 619.15 km2. As a pioneering province in China’s national policy of returning farmland to forests and grasslands, Sichuan Province played a leading role in implementation. Yibin City, located at the confluence of the Jinsha, Min, and Yangtze Rivers, became a key area for the program due to its strategic geographic position. Between 1999 and 2020, Yibin was assigned a reforestation target of 1,297.782 km2. By 2021, this target had been fully achieved through a combination of measures, including converting farmland to forest, afforestation of barren hills and wastelands, and forest closure for regeneration. This policy has had a significant impact on the structural changes between cultivated and forest land within the basin.

Comparison with existing studies

In the Minjiang River Basin, the total cultivated land from 1990 to 2020 exhibited a fluctuating trend of decrease, increase, and subsequent decrease in response to population growth. Conversely, per capita cultivated land did not mirror this trend and consistently decreased over the same period. By 2020, the total cultivated land in the Minjiang River Basin was recorded at 13,133.38 km2, with a per capita value of 0.0005 km2, significantly lower than the national average of 0.000845 km2 per capita. Researchers analyzed the spatiotemporal patterns of cultivated land in the Huaihe, Yangtze, Yellow, and Haihe River Basins within Henan Province (Fan & Liu, 2014). The study revealed that the cultivated land area in the Huaihe River Basin, Yellow River Basin, and Haihe River Basin initial decreased and then increased, while the Yangtze River Basin experienced a gradual increase. Despite these varying trends, the per capita cultivated land area in all four basins consistently decrease, notably lower than the national average of 0.000845 km2 per capita.

The drivers of non-agriculturalization exhibit regional variations due to differing geographical conditions and policies. For instance, researchers conducted an in-depth analysis of the spatial and temporal evolution characteristics of non-agriculturalization and its drivers from 2000 to 2020 in three regions: Jilin Province in Northeast China, Henan Province in the North China Plain, and the Guangdong-Hong Kong-Macao Greater Bay Area on the South Coat (Zhang et al., 2024). Their study unveiled clear geographic variability and temporal volatility in the non-agriculturalization process, closely related to the development strategies, economic conditions, policy environment, and natural environment. Across these regions, socio-economic factors have a significantly greater impact on cropland non-agriculturalization than natural factors. Notably, population growth, agricultural mechanization, and regional transportation are the main drivers of non-agriculturalization. This is also consistent with the findings of this article. The Minjiang River Basin is situated in the western region of China. In a related investigation, Chen, Li & Xia (2022) used Geographic Detector and found that economic development is the main factor influencing non-agriculturalization in the southwestern China, aligning with the conclusions drawn in this study.

This study specifically focuses on the middle and lower reaches of the Minjiang River Basin. The upper reaches of the Minjiang River Basin consist of hilly and mountainous areas with higher elevations, primarily covered by forests and grassland ecosystems, with only a small amount of cropland. In contrast, cropland is predominantly situated in the plains of the middle and lower reaches of the Minjiang River. Researchers consolidated datasets from four subregions of the North China Plain Area to collect information on cropland areas in each subregion (Wei et al., 2019). They observed from the late 17th century to the 1980s, the distribution of cropland expanded to hilly areas, with a continuous increase in cropland rates in plain areas. Notably, cropland in the plain areas experienced a more significant increase compared to that in hilly and mountainous areas.

Non-agriculturalization and non-grainization

In this study, “non-agriculturalization” pertains to the conversion of cultivated land for urban and industrial purposes, as previously defined. It primarily focuses on the change in land use from agricultural to non-agricultural use. The concept of non-grainization, referring to the use of cultivated land for agricultural purposes other than staple grain production like rice, wheat, and corn (Lu et al., 2024). This includes the cultivation of fruits, vegetables, tea, and other economic crops. Although both non-agriculturalization and non-grainization involve changes in land use, non-agriculturalization represents a more radical transformation from agricultural to non-agriculturalization sectors, while non-grainization remains within the agricultural domain but shifts the focus of agricultural production.

It is crucial to highlight that non-grainization can also have implications for food security. When a significant portion of cultivated land is allocated to non-grain crops, the production of staple grains may be adversely affected (Ten Brink, Giesen & Knoope, 2023). Non-agriculturalization directly reduces the amount of cultivated land, while non-grainization diminishes the area dedicated to grain cultivation. Excessive non-agriculturalization or non-grainization not only undermines the rational utilization of cultivated land resources, but also threatens the stability of grain production.

In the Minjiang River Basin, with the development of modern specialty agricultural industries, it is essential to balance the growth of non-grain crops with the protection of staple grain production to ensure food security at both local and national levels. Further investigation into non-grainization in the Minjiang River Basin presents a promising avenue for future research.

Policy inspirations

The study reveals that population growth and the demand for economic development are the main driving factors behind non-agriculturalization. Based on the terrain of the study area and existing researches, we have gained some inspirations that may be of reference value for the land management in the area.

Although the Minjiang River Basin is generally flat, its terrain is not a simple plain. It is also characterized by widely distributed gently rolling earth mounds. Yang used Earth observation and socio-economic data to systematically analyze the expansion of human activities in the Asian Highlands from 2000 to 2020. The study found that 80% of human activity expansion in the region was related to the development of cultivated land, while artificial land use expansion accounted for about 20% (Yang et al., 2022). This research provides important insights for optimizing land use in the Min River Basin. Based on these findings, several strategies can be considered to enhance the development and utilization of the basin’s earth mounds.

On the one hand, these areas serve as valuable reserves for cultivated land. Over the past 30 years, despite facing significant challenges related to the non-agriculturalization of cultivated land (Fig. 7), Chengdu has made notable progress in supplementing cultivated land in the shallow hilly areas of the western Chengdu Plain agricultural zone and the eastern hilly agricultural zone of Longquan Mountain. By 2025, the total cultivated land area in the Chengdu Plain is expected to reach 4,780.69 km2, accounting for 35.90% of the total plain area. Drawing from this successful experience, it is recommended that, in the low-lying hilly areas of the Min River Basin, an assessment of the potential for cultivated land development be conducted, considering natural factors such as soil conditions and water availability. This approach would support the moderate expansion of eco-friendly cultivated land, ensuring food security while preventing ecological degradation.

On the other hand, the interwoven plains and mounds provide favorable conditions for the rational planning of industrial zones. Promoting the development of cities and industrial areas toward adjacent hilly regions holds significant potentials. For example, in Cuiping District of Yibin City, the Minjiang New Area was planned for urban development in mountainous terrain in 2014, with a total development area of 19.388 km2. During the construction of the Minjiang New Area from 2015 to 2020, the proportion of cultivated land in Cuiping District declined only slightly from 74% to 73%. In contrast, prior to the project’s initiation, the cultivated land proportion in this area had been decreasing by an average of 5% every five years. This indicates that urban development towards hilly and mountainous terrain significantly reduces the demand of high-quality farmland, providing a replicable model for balancing urban–rural development and farmland protection in the Min River Basin.

The Minjiang River Basin has many low hilly areas, with substantial development potential. For instance, in Meishan City, the hilly areas of the Longquan and Zonggang mountain ranges cover 4,237.75 km2, accounting for 58.97% of the city’s total area. Through scientific planning and the rational development of these hilly and mountainous resources, the pressure on cultivated land protection in the plains will be effectively alleviated, providing robust support for curbing the non-agriculturalization.

Uncertainties and limitations

(1) Issues about temporal and spatial resolutions: the study on the non-agriculturalization covers a wide temporal scope, with a temporal resolution of 5 years. However, significant changes in cropland can occur within shorter periods. For instance, there may be a substantial increase in cropland area within a specific five-year interval, followed by a notable decrease in subsequent years. This is particularly relevant for crops like wheat, which have a growth cycle lasting from 220 to 270 days within a year, making it impractical to conduct off-farm testing. However, a mere five-year dataset may not yield detailed insights into non-agricultural changes. Analyzing changes over just one year requires more stringent accuracy standards, especially when using a 30 m resolution dataset, which may not precisely capture changes between adjacent years. Thus, employing a 30 m resolution dataset over a five-year period is reasonable in this study. Due to dataset accuracy concerns, this study serves as an introductory exploration of non-agriculturalization issues. Future research on non-grainization will necessitate higher temporal and spatial resolutions for a comprehensive analysis. For instance, it is essential to differentiate between staple crops and other plants, such as fruits and teas, which require data with higher spatial and temporal resolutions (Wang & Xu, 2017).

(2) Issues about driving factors: the drivers of non-agriculturalization are influenced by a limited set of factors. This study excludes the years 1990–1995 from its analysis due to unavailability of data. We acknowledge that there is an issue with missing data; however, based on the analysis of the driving factors in the subsequent years, this has not significantly affected the results. Geodetectors are utilized to examine the driving mechanisms by identifying the spatial distribution of data. If the spatial distribution of changes in the x data within a specific region reflects changes in the y data (the area of non-agriculturalization), the x factor is considered to have explanatory power for cropland denudation (Wang & Xu, 2017).

Findings indicate that over 90% of the driving factors demonstrate high explanatory power for the periods 1995–2000 and 2000–2005. However, the five highest values of factor q for the periods 2005–2010, 2010–2015, and 2015–2020 fall between 0.4 and 0.6, suggesting some randomness in the outcomes.

The study primarily relies on the statistical yearbook to explore driving mechanism, but it recognizes limitations in data accuracy and indicator availability. The study employed administrative regions as units while concentrating on watersheds as the study area. Although there is a degree of spatial mismatch between administrative units and basin units, the core principle remains applicable to research at the basin scale. Prior to conducting the investigation into the drivers of non-agriculturalization, we spatially superimposed the administrative regional data with the river basin to minimize errors resulting from spatial mismatch as much as possible. Although both regions being characterized by intensive human activities, potential data biases are conceivable. Furthermore, the selected indicators in the study may necessitate further refinement. For example, researchers utilized indicators such as soil quality and labor mobility to explore the driving mechanism of the non-agriculturalization of cultivated land in Chinese counties (Jie et al., 2023).

Conclusion

The non-agriculturalization of cultivated land is a major challenge in the context of economic development both in China and globally. This is an exploratory study of non-agriculturalization issues, which conducts a detailed analysis of the non-agriculturalization process and its characteristics in a densely populated area from 1990 to 2020, the Minjiang River Basin. The results indicate that: (1) cultivated land in the Minjiang River Basin was mainly concentrated in the central and southeastern flat areas, with the proportion of cultivated land decreasing from 66.71% to 59.57% over the period; (2) the spatial distribution of cultivated land remained consistent, predominantly in the central and southeastern regions; and (3) from 1995 to 2020, the growths of population and economic were identified as the main drivers of non-agriculturalization in the basin. These research outcomes provide a scientific basis for local governments to devise targeted cultivated land protection policies. For instance, given that population growth and economic development are the primary drivers of non-agriculturalization, the government can strategically adjust population distribution planning and industrial development policies.

A five-year temporal resolution and a 30-meter spatial resolution might be insufficient to capture all changes in cultivated land and crops. However, if further non-grainization of cultivated land is to be studied, data with higher spatial and temporal resolutions will be necessary. The integration of these results with land use change models is recommended. For instance, we can utilize the FLUS model (Liu, Zhao & Song, 2017) to simulate the future non-agriculturalization process in the Minjiang River Basin. Additionally, the coupling of the FLUS model with the InVEST method is a potential solution to investigate future land use evolution and assess its effect on habitat quality (Chen, Zhang & Lin, 2025).

We sincerely thank the editors and reviewers for their valuable suggestions to improve the quality of the article.

Additional Information and Declarations

Competing Interests

Author Contributions

Data Availability

Jingzhe Wang is an Academic Editor of PeerJ.

Xi Zhao conceived and designed the experiments, performed the experiments, analyzed the data, prepared figures and/or tables, authored or reviewed drafts of the article, and approved the final draft.

Zhongwen Hu conceived and designed the experiments, analyzed the data, prepared figures and/or tables, authored or reviewed drafts of the article, and approved the final draft.

Yinghui Zhang conceived and designed the experiments, performed the experiments, analyzed the data, prepared figures and/or tables, authored or reviewed drafts of the article, and approved the final draft.

Jingzhe Wang analyzed the data, authored or reviewed drafts of the article, and approved the final draft.

Tiezhu Shi analyzed the data, authored or reviewed drafts of the article, and approved the final draft.

Yanguo Liu conceived and designed the experiments, analyzed the data, authored or reviewed drafts of the article, and approved the final draft.

Jie Zhang analyzed the data, authored or reviewed drafts of the article, and approved the final draft.

Guofeng Wu analyzed the data, authored or reviewed drafts of the article, and approved the final draft.

The following information was supplied regarding data availability:

The China Multiperiod Land Use Land Cover Change (CNLUCC) remote sensing monitoring dataset is available at DOI: 10.12078/2018070201.

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
