# Peer review of "Non-agriculturalization of cultivated land in densely populated areas at the watershed scale: a case study of the Minjiang River Basin, China"

_PeerJ, doi:10.7717/peerj.19722_

## Round 0.1 · original submission · Major Revisions

· Academic Editor

Major Revisions

Dear Authors,

Thank you for submitting your manuscript entitled “Non-agriculturalization of cultivated land in densely populated areas at the watershed scale: A case study of the Minjiang River Basin, China” to PeerJ.

After a careful evaluation of the manuscript and the reviewers' comments, we believe that your work addresses a relevant and timely topic. However, in its current form, the manuscript requires major revisions before it can be considered for publication.

In particular, the following aspects must be addressed:

Alignment with PeerJ’s Guidelines: Please ensure that the structure, language, and formatting of your manuscript strictly adhere to the journal's submission guidelines. This includes consistency in section headings, proper citation formatting, and clarity in the presentation of results and discussion.

Methodology Description: To enhance clarity, we strongly recommend including a comprehensive diagram or flowchart that visually illustrates the entire methodological workflow of your study.

We encourage you to revise your manuscript thoroughly, taking into consideration both the reviewers' feedback and the journal’s editorial standards. Please provide a detailed response letter highlighting how each comment has been addressed.

We appreciate your interest in PeerJ and look forward to receiving your revised submission.

Best regards,
Armando Sunny

Reviewer 1 ·

Basic reporting

Review of manuscript "Non-agriculturalization of cultivated land in densely populated areas at the watershed scale: A case study of the Minjiang River Basin, China" (peerj-reviewing-113448)
This manuscript tries to examine the characteristics and trends in the spatiotemporal evolution of cultivated land in the Minjiang River Basin, along with the drivers of non-agriculturalization. From 1990 to 2020, the cultivated land in the Minjiang River Basin has decreased and mainly concentrated in flat areas. After reading this manuscript, I have some comments and suggestions as shown in the following.

Experimental design

- . The manuscript points out that the statistical data of some cities from 1990 to 1995 are missing, which makes it impossible to analyze the driving factors during this period, weakening the integrity of the time series analysis. In addition, the study uses a 5-year time resolution (such as 1990-1995, 1995-2000), which may not capture the short-term fluctuations of cultivated land changes, but this is not fully discussed.
- . The spatial resolution of the land use dataset used in this study is 30 meters, which may not be able to distinguish the internal differences of cultivated land (such as the conversion of food crops and cash crops). The manuscript does not explore its impact on the classification of non-agricultural types in depth.
- . The geographic detector relies on spatial heterogeneity, but the manuscript does not clearly state whether it is suitable for the study of non-agricultural driving mechanisms at the watershed scale. For example, the spatial mismatch between administrative units and watershed units is not discussed.
- . The literature review is relatively old, lacking cutting-edge research on non-agriculturalization in the past two years. Please refer to: Investigating the Spatial Distribution and Influencing Factors of Non-Grain Production of Farmland in South China Based on MaxEnt Modeling and Multisource Earth Observation Data.
- . The explanatory power (q-value) of some driving factors from 2005 to 2020 is between 0.4 and 0.6, which shows a certain randomness, but the reasons are not further analyzed.

Validity of the findings

- . Although the trend of northward migration of cultivated land is described by the standard deviation ellipse (SDE), the physical mechanism of migration (such as why it migrates northward instead of other directions) is not explained by combining background factors such as topography and policy.
- . The policy recommendations proposed in the manuscript are relatively generalized and lack specific measures for the characteristics of the study area. More specific suggestions are needed on how to implement effective farmland protection and management policies locally.
- . The concepts of "non-agriculturalization" and "non-grainization" are mixed in the manuscript (Section 4.3), and the difference and connection between the two are not clearly defined, which is easy to cause logical confusion.

Additional comments

- . Table 1 (Distribution density values of cultivated land plots) does not specify the units of “aggregate” and “average”, which may cause misunderstanding.
- . In future studies, the authors can combine the results with land use change models. Please refer to: Projecting future land use evolution and its effect on spatiotemporal patterns of habitat quality in China.
- . Figure 5 (Spatial aggregation degree of non-agriculturalization) does not indicate the specific meaning of the color scale, such as the numerical range corresponding to the density level.
- . The statement in the conclusion section is a bit general and fails to fully highlight the importance and practical significance of the study. In addition, the conclusion section also fails to clearly point out the limitations of the study and the direction of future research.

Reviewer 2 ·

Basic reporting

• Professional English language used throughout.
• I recommend structuring your introduction from general to specific, avoiding constant shifts between these two levels. This approach will enhance the clarity of the problem statement and the justification of the study.
• The figures are significant, but some aspects need to be improved, as detailed below.

Line 37. I suggest you change “non-agriculturalization of cultivated land” as keyword because this word is in the title.
Line 44. I suggest you write FAO.
Line 45 – 52. I suggest adding references to support this information.
Line 53. I recommend that you make the change for “Many researchers”.
Line 54. I suggest adding examples of the factors that influence land non-agriculturalization.
Line 57 – 60. I consider this information was mentioned in the lines 45 – 42 with other word but it was mentioned. I suggest adding this references in line 45 – 52.
Line 68 – 70. I recommend that you make the change for “The basin is also grappling with limited cultivated land resources due to a high population density, uneven water resource distribution, imbalanced economic development, and significant human-land conflicts”.
Line 85. I suggest adding annual precipitation in mm.
Line 94. For Figure 1, my suggestion is to add the necessary map elements. Also, I recommend marking the study area with a dot on the reference map.
Line 98. I suggest mentioning all relevant data used.
Line 105. I recommend that you make the change for “Google Earth Engine (GEE)” and “Digital Elevation Model (DEM)”.
Line 108. I recommend that you make the change for “Average Relative Humidity”.
Line 118, 130, 137, 143, 154, 159. I suggest reviewing the citation method in the author's guide.
Line 125. I recommend that you make the change for “barren soil”, I understand it is bare soil.
Line 136 – 137. It is recommended that the wording be improved, because “this study” is repeated.
Line 164. Please explain why you used five levels in the KDE. I recommend adding this information to this section. I suggest mentioning the advantage of using five levels as Brandes et al. (2024) mentioned in their studies.
Line 167. For Figure 2, please add the unit of measurement to the legend of the map.
Line 168. I suggest taking care not to separate table names and quantities.
Line 197. I recommend capitalizing the first letter in the first letter of attribute names on the tables.
Line 198. For Figure 3, it is important to specify which is Figure 3a and which is Figure 3b. Please add the unit of measurement to the legend of the map.
Line 221. Figure 4 was not mentioned in the text.
Line 225. It is my recommendation that Figure 5 be placed immediately after its presentation in the text.
Line 227. Please specify which figures show a trend of outward expansion from the center of the main urban area.
Line 240. For Figure 5, please add the unit of measurement to the legend of the map.
Line 243. It is my recommendation that Table 4 be placed immediately after its presentation in the text.
Line 272. For Table 4, I recommend change from “Moran′s” to “Moran's”.
Line 273. Figure 6 was not mentioned in the text.
Line 277. I recommend mentioning that in Table 5 in the column “Expounds” the importance of the selected factors is mentioned. Also, I suggest changing “Expounds” to “Importance”.
Line 281. Table 5 was not mentioned in the text.
Line 283. Figure 7 was not mentioned in the text.
Line 301 – 314. I consider it is important to mention in which Figure show these results.
Line 303. Please explain what GDP is because it is not mentioned before. I suggest writing Gross Regional Product (GDP) at the first mention.
Line 318. Figure 8 was not mentioned in the text.
Line 352, 353, 358. Set the superscript to km2
Line 352. I suggest adding the comma to the amount 13133.38
Line 360, 369, 375, 417. I suggest reviewing the citation method in the author's guide.
Line 380 – 394. I recommend adding references to provide support.
Line 402. Please explain what GDP is because it is not mentioned before.
Line 406 – 409. I recommend adding references to provide support.
References
• Brandes, G., Sieg, C., Sander, M., & Henze, R. (2024). Driving Domain Classification Based on Kernel Density Estimation of Urban Land Use and Road Network Scaling Models. Urban Science, 8(2), 48.

Experimental design

I recommend including a diagram of the methodology. This will help you quickly visualize the methods used and to ensure that an international audience can clearly understand your methodology.

Validity of the findings

There is no evidence that the results have been verified.

Additional comments

I suggest reviewing the citation method in the author's guide. Cite, for example, for three or fewer authors, list all author names (e.g. Smith, Jones & Johnson, 2004). For four or more, abbreviate with “first author” et al. (e.g. Smith et al., 2005).

---

## Round 0.2 · Minor Revisions

· Academic Editor

Minor Revisions

Dear Authors,

Thank you for submitting the revised version of your manuscript entitled
“Non-agriculturalization of cultivated land in densely populated areas at the watershed scale: A case study of the Minjiang River Basin, China.” Both reviewers agree that you have done a commendable job incorporating their earlier comments. The manuscript is significantly improved, and the overall structure, data presentation, and discussion have been strengthened.

However, before we can proceed to final acceptance, a few minor corrections remain to be addressed. Below, please find a summary of the outstanding points. Once you have incorporated these revisions, the manuscript should be acceptable for publication.

Thank you for your attention to these details. We appreciate your efforts to improve the quality of this work and look forward to receiving the revised version.

Best regards,

Armando Sunny

Reviewer 1 ·

Basic reporting

no comment

Experimental design

no comment

Validity of the findings

no comment

Additional comments

The authors did a good job on incorporating the reviewers' comments in their revision. The manuscript is significantly improved. I have no further comments.

Reviewer 2 ·

Basic reporting

Line 64 – 72. I suggest adding references to support this information.
Line 133. I suggest adding a more recent reference.
Line 197. I suggest checking the Table number.
Line 232. I recommend that you make the change for “dispersed”.
Line 257. I recommend that you make the change for “As shown in the Figure 6”.
Line 261. I recommend that you make a change for “intensive non-agriculturalization had intensively occurred”.
Line 382. I suggest checking the text quote, Fan and Liu (2024).
Line 441. I recommend adding “an assessment of the potential”.

Experimental design

The authors have included a diagram of methodology. This allows you to view quickly the methods used.

Validity of the findings

No comments.

Additional comments

First, I would like to recognize the authors have done a great job. The manuscript has changed considerably. The authors have answered the questions raised in the first review and have defended their position with arguments. I only have some important observations about conclusions. It is my recommendation that the conclusions of this study include the statement that it is an exploratory study of non-agriculturalization issues.

Annotated reviews are not available for download in order to protect the identity of reviewers who chose to remain anonymous.

---

## Round 0.3 · accepted · Accept

· Academic Editor

Accept

Dear Authors,

Thank you for your revised submission of the manuscript entitled:

"Non-agriculturalization of cultivated land in densely populated areas at the watershed scale: A case study of the Minjiang River Basin, China"

I have now received and reviewed the final comments from both reviewers. I am pleased to inform you that both reviewers agree that you have done a commendable job incorporating their earlier suggestions and concerns, and they are satisfied with the revisions you have made.

In light of the positive evaluations and the quality of the revised manuscript, I am happy to inform you that your article is accepted for publication in PeerJ.

Thank you for submitting your work to PeerJ. We look forward to seeing your article published and shared with the broader scientific community.

Best regards,
Dr. Armando Sunny

Reviewer 2 ·

Basic reporting

The authors have answered the questions raised in the two reviews, and have defended their position with arguments. These arguments are convincing and justified.

Experimental design

The authors have answered the questions raised in the two reviews, and have defended their position with arguments. These arguments are convincing and justified

Validity of the findings

The authors have answered the questions raised in the two reviews, and have defended their position with arguments. These arguments are convincing and justified

Additional comments

The article should be accepted.